# The economic costs of planting, preserving, and managing the world's forests to mitigate climate change

K. G. Austin [1✉], J. S. Baker [1,2], B. L. Sohngen[3], C. M. Wade[1], A. Daigneault[4], S. B. Ohrel [5], S. Ragnauth [5] & A. Bean[1]

Forests are critical for stabilizing our climate, but costs of mitigation over space, time, and stakeholder group remain uncertain. Using the Global Timber Model, we project mitigation potential and costs for four abatement activities across 16 regions for carbon price scenarios of \$5–\$100/tCO$_2$. We project 0.6–6.0 GtCO$_2$ yr$^{-1}$ in global mitigation by 2055 at costs of 2–393 billion USD yr$^{-1}$, with avoided tropical deforestation comprising 30–54% of total mitigation. Higher prices incentivize larger mitigation proportions via rotation and forest management activities in temperate and boreal biomes. Forest area increases 415–875 Mha relative to the baseline by 2055 at prices \$35–\$100/tCO$_2$, with intensive plantations comprising <7% of this increase. Mitigation costs borne by private land managers comprise less than one-quarter of total costs. For forests to contribute ~10% of mitigation needed to limit global warming to 1.5 °C, carbon prices will need to reach \$281/tCO$_2$ in 2055.

[1] RTI International, 3040 E Cornwallis Rd, Durham, NC 27709, USA. [2] Department of Forestry and Environmental Resources, North Carolina State University, 2800 Faucette Dr, Raleigh, NC 27607, USA. [3] Department of Agricultural, Environmental and Development Economics, The Ohio State University, Columbus, OH 43210, USA. [4] School of Forest Resources, University of Maine, Orono, ME 04469, USA. [5] US EPA, 1200 Pennsylvania Avenue, N.W, Washington, DC 20460, USA. ✉email: kaustin@rti.org

It has long been recognized that nature-based climate change mitigation strategies, which leverage natural ecosystem carbon sequestration and storage processes, have the potential to substantially reduce global greenhouse gas (GHG) emissions[1,2]. The Intergovernmental Panel on Climate Change (IPCC) reports that such strategies, which include afforestation, reforestation, improved forest management, and avoided forest conversion, can play a critical role in the global abatement portfolio[3], and recent IPCC special reports highlight the need for land-based mitigation to prevent warming above 1.5–2 °C[4,5]. Forest sector activities are also prominent in national and international commitments to mitigate GHG emissions[6–8]. Previous research suggests that these abatement activities have the potential to achieve one-quarter to one-third of the mitigation required to meet climate stabilization targets by 2030 (refs. [9–18]).

We investigate the cost of reducing global GHG emissions via forest sector abatement activities using the Global Timber Model (GTM), a dynamic economic model of the global forest sector[19–21]. Using GTM allows us to capture endogenous feedbacks between markets and land use, intertemporal trade-offs, interactions between abatement actions, and spatial allocation in harvest patterns and mitigation activities, all of which have a strong influence on abatement costs, and which have not been previously considered in estimates of the cost of global forest sector mitigation[22]. The ability to capture these processes is critical, as proposed mitigation policies will have strong implications for markets and do not have a historical precedent[23]. Furthermore, we build on previous empirical research to consider not just potential net returns from alternative land uses (e.g., avoided deforestation, reforestation)[24] but also the changing marginal user costs of forest management (e.g., increasing harvest rotations, changing fertilizer application, species selection, or seedling density).

We used GTM to establish a reference level of future GHG fluxes from land and forest management in the absence of mitigation incentives, calibrated to macroeconomic projections of GDP and global population from the Shared Socioeconomic Pathways (SSP) scenario 2 (ref. [25]), and the US Annual Energy Outlook[26]. We compared this reference case to scenarios in which GHG mitigation incentives are provided in the form of rental payments for carbon sequestration. We considered a range of assumed prices ($5, $20, $35, $50, $75, and $100/tCO_2) and both low and high price growth rates (1 and 3% per year). Our analysis projects GHG emission reductions in response to mitigation price signals for four forest sector mitigation activities: avoided deforestation, forest management activities, increasing harvest rotations, and afforestation/reforestation (including natural forest regeneration and intensively managed timber plantation establishment). We project the total quantity and cost of mitigation for 16 global regions, disaggregate mitigation results by activity and country, and illustrate how mitigation portfolios vary over time. We report mitigation quantities and costs in the near term (2015–2035) and out to mid-century (2015–2055).

In addition, we differentiate the cost of mitigation incurred by private land managers, which includes the costs of carbon storage via forest management, harvest, and maintaining or increasing forest cover, from the total costs. The total cost metric assumes every unit of carbon sequestration additional to the baseline receives the same price incentive. Land managers who face a low marginal or opportunity cost of increasing carbon storage relative to the price incentive would thus earn excess economic rents for participating in the program and storing additional carbon. Understanding the difference between the total cost of mitigation and the costs incurred by land managers (including possible benefits of higher output prices) is useful for identifying cost-reducing policy strategies that cover land manager abatement costs (including opportunity costs of delayed harvests) while reducing the total cost of emissions reductions to society.

## Results

**Global forest sector mitigation potential and costs**. Our projections range from modest mitigation at low carbon price scenarios to more ambitious mitigation consistent with those reported by the IPCC at high prices[27] (Table 1, Supplementary Fig. 1 and Supplementary Data 1). Across mitigation price scenarios we project cumulative GHG mitigation in the global forest sector of 12.1–102.9 GtCO_2 by 2035. This is equivalent to an average annual mitigation of 0.6–5.2 GtCO_2 yr$^{-1}$ over the 2015–2035 period, at a total annual cost of 1–178 billion USD yr$^{-1}$. By 2055, we project cumulative mitigation of 25.2–239.1 GtCO_2, or average annual mitigation of 0.6–6.0 GtCO_2 yr$^{-1}$ at a cost of 2–393 billion USD yr$^{-1}$. The large range of potential, as well as costs, illustrates that forests can be a source of mitigation effort under a range of carbon prices. However, as has been shown across other sectors of the economy[28], marginal abatement costs are steep above certain price thresholds, with diminishing returns for increasing mitigation incentives (Fig. 1). For example, a scenario that provides mitigation of 2.1 GtCO_2 yr$^{-1}$ costs $16 billion yr$^{-1}$ in 2035, while roughly doubling this rate of mitigation to 4.6 GtCO_2 yr$^{-1}$ costs $116 billion yr$^{-1}$.

We project that avoided deforestation results in average annual mitigation of 0.3–1.8 GtCO_2 yr$^{-1}$ over 2025–2055, while afforestation and reforestation results in 0.1–2.6 GtCO_2 yr$^{-1}$ and forest management including changes in harvest rotations results in 0.2–1.6 GtCO_2 yr$^{-1}$. These activity-specific mitigation projections fall on the low end of the ranges reported by Roe et al.[29] from the literature, not limited to those studies which estimate the economically efficient level of mitigation. This review reports that reducing deforestation and forest degradation could result in mitigation of 0.4–5.8 and 1.0–2.2 GtCO_2 yr$^{-1}$ over 2020–2050, respectively. Further, it provides estimates of the range of mitigation achievable via afforestation/reforestation of 0.5–10.1 GtCO_2 yr$^{-1}$ and of forest management of 0.4–2.1 GtCO_2 yr$^{-1}$ over the same period. We do not include projections that result in mitigation quantities at the higher end of this reported range by mid-century, but project that achieving this level of mitigation by the year 2100 would require carbon prices to exceed $1000/tCO_2.

We project that the cost of emissions reductions in the forest sector is higher than previously reported in recent literature. For example, Griscom et al.[22] reported 'cost effective' mitigation potential in the forest sector of roughly 7.0 GtCO_2 yr$^{-1}$ by 2030 at carbon prices less than $100/tCO_2, while we project average mitigation potential of 5.2 GtCO_2 yr$^{-1}$ at an initial carbon price of $100/tCO_2 assuming 1% growth. Busch et al.[17] reported cumulative mitigation potential from tropical reforestation and avoided deforestation of 60.8 GtCO_2 at a price of $20/tCO_2, and 123.4 GtCO_2 at a price of $50/tCO_2, by 2050. We project similar levels of cumulative mitigation in the tropics of 61.7 GtCO_2 and 107.7 GtCO_2 by 2055 under initial carbon prices of $20/tCO_2 and $50/tCO_2, respectively, at 1% growth. However, our approach assumes that these initial prices increase over time, resulting in prices of just under $30/tCO_2 and $75/tCO_2 in 2055. Our higher projected cost estimates relative to these previous studies are due to GTM's representation of market interactions between regions, intertemporal dynamics, and endogenous land management responses to policy signals[30].

**Distribution of projected forest sector mitigation**. The largest share of global forest sector mitigation will come from the tropics, where 72–82% of total global mitigation will occur across all

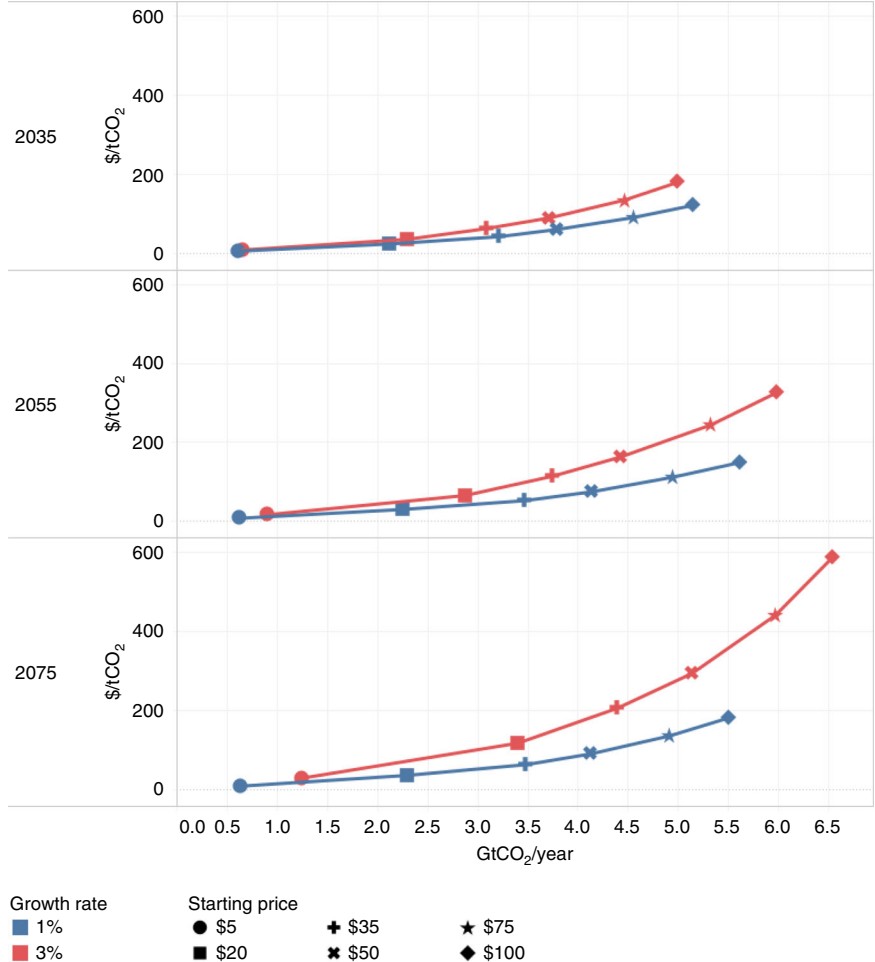

**Fig. 1 Global marginal abatement cost curves in the global forest sector.** We project mitigation quantities and costs across three time-horizons: 2035 (top), 2055 (middle), and 2075 (bottom), under six starting prices and two growth rate assumptions.

scenarios in 2055 (Fig. 2). Avoided deforestation and re/afforestation in the tropics remain the dominant sources of mitigation over the study period, comprising 82–86% of mitigation in the tropics in 2055, and 61–71% of total global mitigation in 2055. The contribution of afforestation/reforestation diminishes somewhat in the tropics in the low-growth scenarios, due to high land opportunity costs for the remaining area that can be forested.

As high levels of avoided deforestation reduce timber supply in the tropics, we project that timber harvest levels will initially increase in the temperate and boreal (Supplementary Data 2). This higher initial level of harvest is followed by replanting to promote longer-term carbon sequestration beginning in mid-century when price incentives are anticipated to be substantially higher. Over time and at higher price incentives, a larger share of total mitigation shifts to rotation and forest management activities in the temperate and boreal biomes. Rotation changes comprise 18–41%, and management activities comprise 12–32%, of mitigation in these biomes across all scenarios by 2055. By mid-century the temperate and boreal biomes contribute 18–28% of total mitigation in the global forestry sector.

We find that the timing of mitigation is sensitive to carbon price growth rate assumptions (Table 1 and Fig. 2). In the temperate and boreal biomes, where mitigation potential is dominated by rotation and forest management decisions, mitigation may be delayed if land managers anticipate rapid growth in mitigation payment incentives, without policies to regulate the quantity of sequestration. As a result, there is up to

10% less cumulative abatement in the first two decades of the simulation horizon under high carbon price growth scenarios, compared to low growth scenarios (Table 1). However, by 2055 the high growth scenarios result in net abatement that is up to 30% higher than under low growth scenarios, indicating that the effect of high carbon price growth does not persist beyond mid-century.

In the tropics, Brazil, the Democratic Republic of Congo (DRC), and Indonesia contribute the largest share of mitigation across all scenarios (Fig. 3). We project that Brazil's contribution to total tropical mitigation will decline from 9–28% in 2035 to 6–18% by 2055. On the other hand, DRC and Indonesia's contribution to tropical mitigation remains relative stable over 2015−2055, at 6–16% and 5–8%, respectively. Outside the tropics, the US is projected to contribute the largest share of mitigation: 24–30% of mitigation in the temperate and boreal biomes combined, by 2055. Mitigation in the US is primarily driven by management intensification and afforestation, rather than changes in rotation lengths. Canada and China also contribute significantly to temperate and boreal region mitigation, contributing 19–22% and 9–19%, respectively, by 2055.

**Changes in forest area under mitigation incentives**. Total forest area increases under all mitigation price scenarios due to reduced deforestation and afforestation/reforestation relative to the baseline scenario (Supplementary Data 3). In the $35/tCO$_2$ cases forest area increases by 327–360 Mha by 2035, on par with the

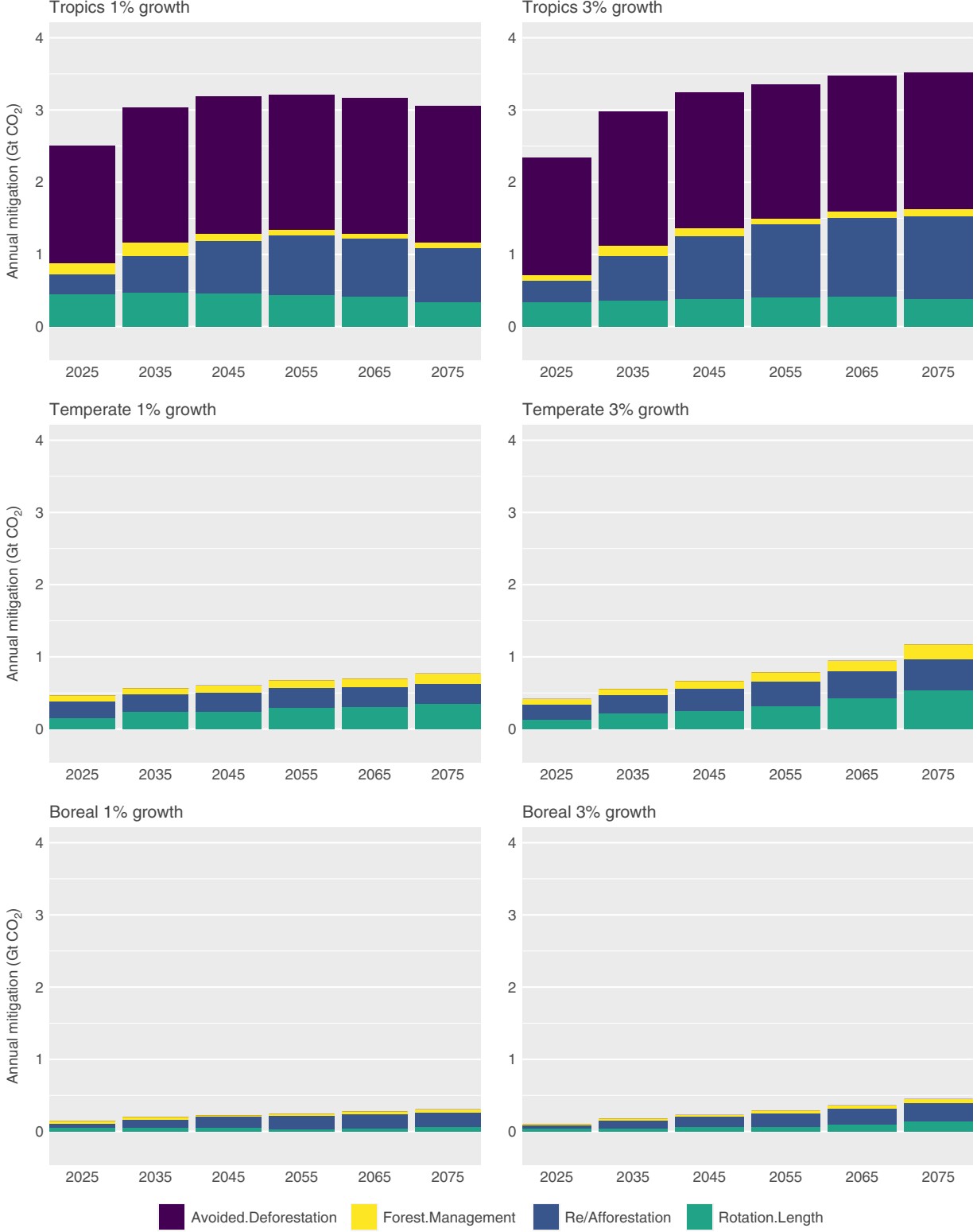

**Fig. 2 Projected annual mitigation across biome, by mitigation activity (avoided deforestation, forest management, re/afforestation, and changes in rotation lengths) at a starting carbon price of $50/tCO$_2$ and under 1 and 3% growth scenarios.** In the temperate and boreal biomes, where forest loss is predominately due to forestry activities, we include carbon gains due to avoided forest in our measure of gains due to shifts in rotation length.

350 Mha of forest restoration pledged under the Bonn Challenge by 2030 (ref. [31]). Under our highest price scenarios of $100/tCO$_2$, forest area could increase by 584–631 Mha by 2035. The largest levels of regional forest cover increase relative to the baseline occur across the tropics with large increases in South America,

Southeast Asia, Sub-Saharan Africa, and in the temperate and boreal regions including in Russia, China, and Canada.

Intensively managed plantations comprise less than 7% of the projected additional increases in forest areas globally under all price scenarios by 2055 (Supplementary Data 3). The proportion

of additional forest area comprising plantations decreases somewhat with increasing carbon prices, as higher price scenarios place more emphasis on conserving existing forests. The relatively small share of intensively managed plantations may alleviate some of the concerns regarding the large-scale conversion of natural ecosystems to monoculture plantations and associated negative impacts on biodiversity and ecosystem-regulating services[32]. However, plantations play a large role in China and the US, particularly under high mitigation price scenarios. For example, plantations make up more than one-quarter of additional forest area under a $100/tCO_2 carbon price in both China and the US, by 2055. Additional plantation investment is one reason why we see harvest levels increase substantially in the US under all mitigation scenarios, and suggests that productive forestry regions like the US could increase forest carbon sequestration while simultaneously recognizing greater comparative advantage in the global forest product market. The net impacts of plantation expansion on biodiversity will depend to a large extent on the prior land cover, and merits further investigation outside the scope of the present study.

**Proportion of mitigation costs borne by land managers**. We isolate the cost of mitigation incurred by private land managers, which include the costs of forest management, changes in harvest patterns, and maintaining or increasing forest cover. The mitigation costs borne by land managers, including individuals, communities that manage forests, and government owners, remain less than one-quarter of total cost of mitigation across all scenarios in 2055 (Table 1). In our lowest price scenarios land managers take advantage of lowest cost abatement actions, or "low hanging fruit", and thus land manager costs are very low or even negative. Producers may see net benefits (negative costs) due to higher timber prices, which occurs when low-cost abatement strategies such as avoided deforestation result in lower harvest levels and price increases that benefit land managers more than the opportunity costs of foregone harvests. At higher carbon prices, land managers invest in more costly abatement actions, and thus the proportion of total mitigation costs borne by land managers increases.

The large difference between the prices paid for mitigation by consumers and third parties and the actual costs incurred by some land managers (including possible benefits of higher output prices) suggests that there is the potential to reduce overall mitigation cost to society with innovative compensation strategies. By better understanding the difference between the total cost

### Table 1 Projected average annual forest sector mitigation and costs.

| | Average annual mitigation (GtCO_2 yr^{-1})[a] | | Annual cost (billion USD yr^{-1})[b] | | Annual land manager cost (billion USD yr^{-1})[c] |
|---|---|---|---|---|---|
| | 2035 | 2055 | 2035 | 2055 | 2055 |
| $5@1% | 0.6 | 0.6 | 1.1 | 1.9 | −0.1 |
| $5@3% | 0.7 | 0.9 | 1.2 | 2.7 | −0.12 |
| $20@1% | 2.1 | 2.3 | 15.7 | 26.9 | 1.21 |
| $20@3% | 2.3 | 2.9 | 17.1 | 36.7 | 5.31 |
| $35@1% | 3.2 | 3.51 | 38.6 | 70.5 | 4.93 |
| $35@3% | 3.1 | 3.7 | 38.8 | 85.4 | 11.5 |
| $50@1% | 3.8 | 4.1 | 65.5 | 120 | 8.73 |
| $50@3% | 3.7 | 4.4 | 66.5 | 145.4 | 17 |
| $75@1% | 4.6 | 4.9 | 115.5 | 216.9 | 16.3 |
| $75@3% | 4.5 | 5.3 | 120.3 | 262.9 | 28.2 |
| $100@1% | 5.2 | 5.6 | 170.7 | 324.9 | 25.1 |
| $100@3% | 5.0 | 6.0 | 177.5 | 392.7 | 40.5 |

[a]Projected average annual mitigation (GtCO_2 yr^{-1}) in the global forest sector, by carbon price/growth scenario and in the years 2035 and 2055.
[b]Projected annual cost of mitigation, by carbon price/growth scenario and in the years 2035 and 2055.
[c]Projected cost of mitigation borne by private land managers, by carbon price/growth scenario and in 2055.

Total annual mitigation (MtCO_2 yr^{-1})

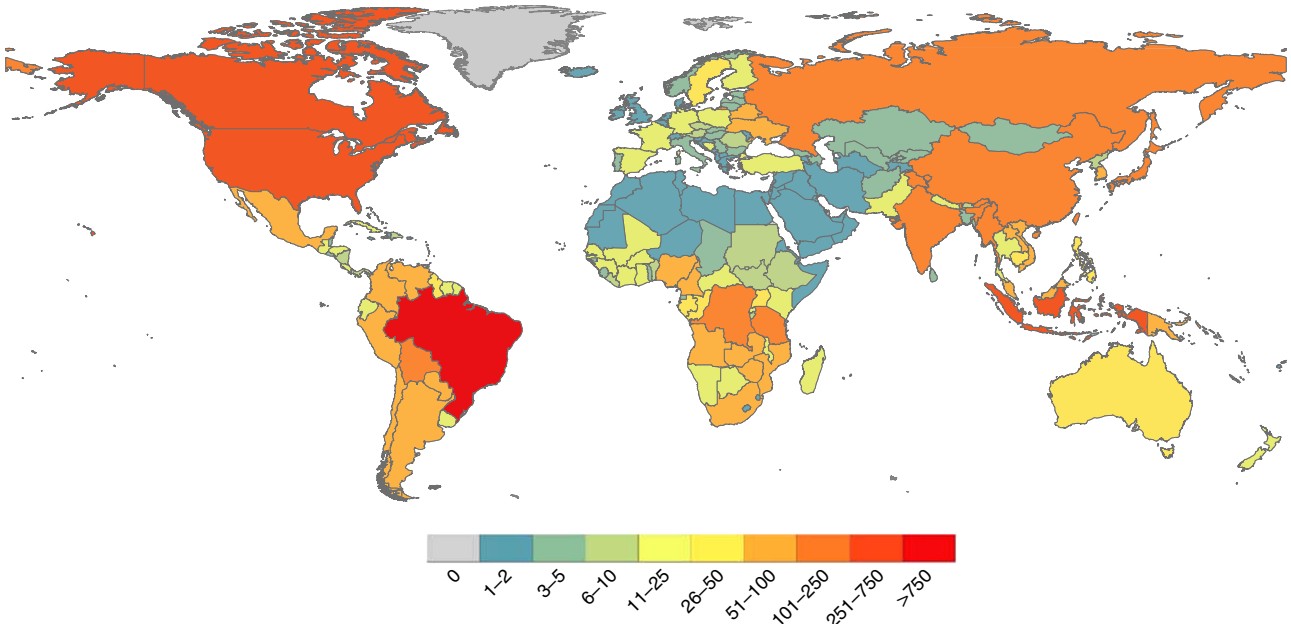

**Fig. 3 Projected annual GHG mitigation by country, across all forest sector abatement activities in 2055.** Mitigation is measured in MtCO_2 yr^{-1} and is presented on a scale from grey (no mitigation potential) to red (largest mitigation potential).

of mitigation and the costs incurred by private land managers (including possible benefits of higher forest product output prices), public entities with budget constraints may be able to design mitigation programs that achieve target mitigation quantities, such as nationally determined contributions (NDCs), at lower cost than under a global carbon market.

## Discussion

This study estimates the economic costs of global forest carbon sequestration commensurate with the 1.5 °C climate stabilization target, using an economic model that accounts for market feedbacks, opportunity costs, intertemporal trade-offs, and interactions between abatement actions. Our resulting mitigation projections of 0.6–5.2 Gt $CO_2$ $yr^{-1}$ by 2035 and 0.6–6.0 $GtCO_2$ $yr^{-1}$ by 2055 thus provides a more accurate reflection of the potential range in average annual mitigation, relative to previous estimates that do not account for these influential dynamics[27]. Future research that improves and expands on this assessment by considering, for example, non-GHG aspects of climate forcing, non-$CO_2$ fluxes, the potential impacts of climate change on forest growth and productivity, and the influence of bioenergy policies, will further refine our understanding of the role of forests in global climate stabilization.

Our results reflect the importance of avoiding deforestation and afforestation/reforestation, long acknowledged to be critical abatement actions for climate stabilization. We project that reducing deforestation results in mitigation of 0.3–1.8 $GtCO_2$ $yr^{-1}$, while afforestation/reforestation sequesters 0.1–2.6 $GtCO_2$ $yr^{-1}$, by 2055. The majority of emissions reductions will occur in the tropics; we estimate that 30–54% of total global mitigation is derived from avoided deforestation in the tropics in 2055, of which 14–38% occurs in Sub Saharan Africa, 10–26% in South America, and 9–14% in Southeast Asia. As high levels of avoided deforestation reduce global timber supply, timber harvests will initially increase in the temperate and boreal, followed by replanting and improved forest management. Thus, the temperate and boreal contribute 15–22% of total mitigation by 2035, increasing to 18–28% in 2055.

Most integrated assessment models (IAM) rely heavily on near-term negative emissions to reach climate stabilization targets, primarily from avoided deforestation, forest expansion, agricultural mitigation, and bioenergy with carbon capture and storage[33,34]. Here, we demonstrate the importance of considering forest management-based mitigation strategies, which can reduce emissions by 0.2–1.6 $GtCO_2$ $yr^{-1}$ by 2055, and contribute up to one-quarter of total forest sector mitigation and more than half of forest sector mitigation in the temperate and boreal biomes. Current IAMs also demonstrate that emissions trajectories over 2020–2030 largely determine the likelihood of limiting warming to 1.5 °C[29]. Our results illustrate that economically efficient forest carbon sequestration pathways are path dependent and could be driven by expectations of future mitigation policies and incentive structures. For example, we find that investments in forest management in the temperate and boreal regions may be delayed in anticipation of higher carbon prices under high growth rate scenarios. Projecting mitigation outcomes with models that account for intertemporal dynamics and endogenous forest management, such as GTM, are critical for demonstrating these path dependencies.

We estimate that investments of 171 billion USD $yr^{-1}$ could result in abatement of 5.2 $GtCO_2$ $yr^{-1}$ in 2035 and 393 billion USD $yr^{-1}$ could result in 6.0 $GtCO_2$ $yr^{-1}$ of abatement in the global forest sector by 2055. This corresponds to 15% and 10% of the level of mitigation needed to limit global warming to 1.5 °C by 2030 and by mid-century, respectively[29]. Notably, the cost of achieving 40% of this level of mitigation is more than 90% lower,

illustrating that marginal costs increase steeply in the sector. Nevertheless, mitigating even 6.0 $GtCO_2$ $yr^{-1}$ in the global forest sector may be possible at a carbon price of \$281/$tCO_2$ in 2055, a value substantially lower than the reported median cost of mitigation in 1.5 °C pathways of \$480/$tCO_2$ across all sectors[35]. Our findings thus confirm a critical role for the global forest sector in achieving cost-effective global climate change mitigation.

## Methods

**GTM framework.** GTM is a dynamic intertemporal economic optimization model representing the global forestry sector that can be used to project forest resource use under assumed future macroeconomic and environmental conditions. The model evaluates four alternative forest and land management options: (1) changing harvest rotation lengths, (2) forest management, (3) afforestation/reforestation, and (4) avoided deforestation, and determines optimal levels of these management interventions over time to maximize the producers' and consumers' surplus (net welfare)[11,21,36,37].

GTM solves all time periods across a 150–200 year timeframe simultaneously via intertemporal optimization. This approach reflects the nature of forestry investments decisions, which are frequently made based on expected returns several decades in the future. Unlike recursive dynamic modeling approaches, which do not consider incentives in the future and will therefore favor investments in activities with near-term mitigation benefits such as avoided deforestation, intertemporal optimization frameworks reflect rational expectations about future market and policy conditions, thus choosing abatement portfolios in response to both current and expected future mitigation prices or policies. Thus, investments in forest management activities or new forests are made by the model in anticipation of higher future mitigation incentives. Intertemporal optimization models assume that landowners and other market actors in the forest sector make forward looking decisions, and that these decisions are not systematically biased (e.g., everyone assumes prices rise 5% per year when they actually rise 2% per year). Intertemporal optimization models, like most of the climate mitigation literature, do not account for uncertainty in future market, policy, or climate trends.

GTM represents 275 forest and management types across 16 regions globally, each interacting through the global timber market. The land classes are differentiated by forest types, accessibility, and management intensity. The forest types are associated with biomes defined in the MC1/2 model[38,39], but are also differentiated by hardwoods, softwoods, or intensively managed plantations, where country-level data exists.

For managed forests, the model determines the rotation age, the intensity of forest management (measured in \$/ha), and the area of land to maintain in managed forests. For inaccessible forests in the temperate and boreal regions, the model determines how many hectares to harvest given a forest access cost function, and forests that are harvested are considered accessible thereafter. Land rents for inaccessible forests in the boreal and temperate regions are assumed to be 0 in the model, meaning these forests remain forests over time, although if they are harvested, they will regrow according to growth functions specific to the forest, land, and management class.

For inaccessible forests in the tropics, the model determines how many hectares to harvest given a forest access cost function and a forest rental function, and how many hectares to maintain in forest (e.g., how many hectares to allow to regenerate). In the tropics, the model assumes that observed land-clearing implies a positive rent for forests at the accessible/inaccessible margin. If the economic productivity (net present value) of keeping an accessed tropical forest is less than the future rents given by the rental function, the forest will be converted to something else. The rental functions are shifted exogenously over time to simulate rising demand for land to be used in agriculture, resulting in land use change. The exogenous factors driving land use change have been calibrated to past land use change and reflect assumptions about future demand for agricultural products by region.

Investments made in GHG mitigation in one region (e.g., delayed harvests or reduced deforestation) will have market implications and can alter harvest and production patterns in other regions, which can induce positive or negative emissions leakage. The current version of the GTM includes the integration of heterogeneous forest product demand functions for various wood products, which allows policy incentives and carbon sequestration investments to influence multiple markets, and which refine projections of regional responses to global stimuli[20,40]. Specifically, this differentiation of forest products allows the model to substitute across pulpwood and sawtimber production to balance forest product demand with demand for forest carbon sequestration under an explicit mitigation policy incentive (i.e., a carbon price). For example, a policy that incentivizes short-term expansion in storage of terrestrial carbon could result in avoided deforestation, near-term expansion of fast-growing plantations, and increased harvest levels of lower-valued pulpwood, with each investment activity concentrated in the most productive regions.

Detailed discussion of the model's underlying data as well as documentation of key parameter sets and structural equations can be found in refs. [20,41,42]. Examples of previous applications of GTM include projecting climate change impacts[19,43,44], forest carbon sequestration potential[11,16,45], and bioenergy production[21,34,46].

Additional detail, and the algebraic structure of the model, can be found in a Supplemental Appendix to the US Mid-Century Strategy for Deep Decarbonization[42]. Details on carbon accounting within GTM are provided in ref. [19].

**Carbon valuation.** Carbon is valued in the model by renting carbon stored in forest stocks, including aboveground carbon, deadwood, slash, and soil carbon. Product market carbon is paid for at harvest time, and is paid the price for the present value of the carbon stored permanently. This is the initial carbon stored less the present value of the path of carbon emissions that occur as forest products decompose in landfills, or are burned. Mathematically, from the perspective of a single stand of trees, carbon rent is as follows:

$$\text{stand value} = \text{Max} \frac{P_S V(T) e^{-rT} - C + \int_0^T R^C(a)\alpha(t)V(a)e^{-ra}\,da + P^C(T)\beta V(T)e^{-rT}}{(1-e^{-rT})},$$

(1)

where

- $P_S$ = price of timber;
- $V(a)$ = yield function (m$^3$ ha$^{-1}$);
- $r$ = interest rate;
- $t$ = time ($T$ = optimal time to harvest trees);
- $C$ = regeneration cost;
- $R^C(t)$ = carbon rent;
- $\alpha(t)$ = biomass expansion factor (tCO$_2$ per m$^3$), adjusted to include C in deadwood, soils, and slash;
- $\beta$ = proportion of harvested wood volume that is stored permanently wood products;
- $P^C(t)$ = carbon price.

$T$, the optimal time to harvest the stand, is determined by maximizing the equation with respect to $t$. Carbon rent is related to carbon price as follows: $R^C(t) = P^C(t) - P^C(t+1)/(1+r)$. As shown in refs. [11,47], this approach properly values the interaction between the carbon in the atmosphere and carbon in forests. This formulation allows for an exact relationship between the current rent and the current carbon price. Furthermore, in efficient markets, as modeled here, the price and the marginal cost are equivalent, so it allows us to determine the marginal cost (i.e., price) of each ton of carbon sequestered in each time period directly.

The GTM can be formulated to value the entire stock of carbon in forests, and the entire stock of carbon stored permanently in wood products, or it can be formulated to value only the additional carbon stored in these pools above the baseline (e.g., additionality). The outcomes of the model are identical in both cases because the decisions made by the model under either set of incentives are the same. The total set of payments under each approach will differ, but the amount of carbon sequestered in each time period will be the same.

Through the rental approach, permanence is not a concern because only the carbon kept out of the atmosphere is incentivized. Carbon that goes into the atmosphere due to disturbance or markets is not incentivized, but the model efficiently manages it. Furthermore, by valuing carbon gains in all forests in the world, the model does not allow forest carbon leakage.

**GTM validation.** Previous studies have demonstrated the ability of the GTM model to accurately project changes in forest resource base, forest management, and harvest levels. Mendelsohn et al.[48] evaluate the performance of GTM in representing historical forest management and land use starting from 1900, by comparing GTM estimates with observed forest land use and harvest levels using the same elasticity assumptions present in the modeling framework. This study reports that GTM performs well and highlights the importance of management's contributions to the historic evolution of forest carbon stocks. Furthermore, Sohngen et al.[41] demonstrate the relative stability in GTM projections under uncertainty in future forest yield growth functions and economic parameters governing the opportunity costs of forest land, increasing confidence that GTM projections are stable relative to observed forest land use, harvests, and market conditions.

We also evaluated the performance of GTM for the 2015–2020 period, by comparing GTM projections of regional harvest levels for industrial roundwood (total less fuelwood), pulpwood, and sawtimber to reported FAO harvest statistics over the same region[49]. We aggregated FAO country level statistics to GTM regions to facilitate comparison. We found that total roundwood harvests are within 12% of reported harvest statistics at the global scale, and that differences are relatively small across most productive timber regions (Supplementary Data 4). This provides confidence in the ability of GTM to project the dynamics of the global forest sector for the purpose of the present analysis.

**GTM scenario design.** We used GTM to first establish a baseline level of future forest and land management, and associated GHG fluxes, in the absence of mitigation incentives. We focused on two key points in the simulation horizon, 2035 and 2055, that also correspond to relevant policy dialog and recent GHG mitigation literature. We calibrated the baseline scenario to macroeconomic projections of

GDP and population worldwide from the SSP scenario 2 (ref. [25]), and for the US from the US Annual Energy Outlook[50]. Macroeconomic projections inform the GTM assumptions regarding future demand growth for pulpwood and sawtimber[19,40]. We then compared this reference case to scenarios in which GHG mitigation incentives are provided in the form of rental payments for carbon sequestration.

Permanent forest loss in temperate and boreal regions has slowed over the past several decades due to demographic transitions, changes in consumption patterns, and greater investments in forest management. Tropical forest loss is likely to decline in the future due to slowing population growth, shifts in consumer preferences, and stricter land use policy[51]. These land use patterns are captured by our reference scenario projections, which is calibrated to FAO forest land use for different regions in the model and through land rental functions that are adjusted over time to reflect higher opportunity costs of both agricultural and forest land.

We considered a range of assumed ambition in land-based mitigation investments by including six potential carbon prices ($5, $20, $35, $50, $75, and $100/tCO$_2$) and both low and high assumed carbon price growth rates (1 and 3% per year). These price scenarios are consistent with the scale of current carbon prices[52] and prices examined in the literature[40,45,53]. The two carbon price growth rates represent differences in the relative value of mitigation over time that may influence land management decisions. These scenarios resulted in projected carbon prices of between $6 and 156/tCO$_2$ by 2035 and $7 and 281/tCO$_2$ by 2055. This scenario design allows us to project the range in potential mitigation, and to isolate the relative importance that initial starting price and growth rates have on projected global and regional mitigation portfolios.

We developed net mitigation projections by comparing the GHG flux under the projected baseline from aboveground biomass, belowground biomass, and soil carbon with the GHG fluxes under the alternative price scenarios from the same carbon pools. By incentivizing the storage of carbon through mitigation price scenarios, carbon storage is operationally treated as a forest product—that is, terrestrial carbon storage can potentially compete with the demand for timber for sawtimber, pulp, and paper products. A price on carbon can result in lower sawtimber market output prices by incentivizing carbon storage in wood products and increasing sawtimber forest stands. We used this difference in carbon storage to estimate the costs of mitigation by multiplying the additional carbon stored in each decade by the average CO$_2$ price for that decade, resulting in the total investment cost metric. We then discounted this investment value at a 5% rate to calculate a net present value and annualized payment for the investment in each mitigation scenario and decadal time step. Annual cost of mitigation is calculated:

$$\text{AnMit}_t = (\text{Mit}_t * P_t)/10,$$

(2)

where $t$ is the decade, Mit$_t$ is decadal mitigation, and $P_t$ is decadal average CO$_2$ price. The NPV of cumulative investment is calculated as

$$\text{NPV}_t = \left(\text{AnMit}_t * \sum_{n=0}^{N-1} (i+r)^n\right) * R + \text{NPV}_{t-1},$$

(3)

where $r$ is the discount rate (5%), $N$ is years in the decade, and $R = (1+r)^{10}$ annualizes the decadal investments.

**Disaggregation by mitigation activity.** The optimization procedure in GTM determines the mix of the four abatement activities and the resulting carbon flux simultaneously under a given carbon price. To determine the contribution of each of these individual activities, we constructed scenarios that allow one activity at a time. We modeled avoided deforestation and afforestation together for the purposes of this exercise, and later decomposed the two effects. To determine the approximate contribution of adjustments in rotation ages to carbon sequestration, we fixed forest area and management intensity to their initial levels. The only change the model could make was in the amount of timber harvested annually, which implicitly adjusts rotation ages. For forest management, we fixed forest area and rotation ages and solved the model. Finally, for avoided deforestation and afforestation, we fixed management and rotation ages, and allowed land area to adjust.

GTM represents forest changes at the extensive margin in one category including both avoided deforestation and afforestation/reforestation. To isolate avoided deforestation we compare the area of forest lost in the baseline to the area lost in each scenario, while conversely we compare the area of forest gained in the baseline to the area gained in each scenario to isolate afforestation/reforestation. However, in the temperate and boreal biomes forest loss is predominantly driven by forestry activities[54] and net forest cover is projected to increase. We therefore include carbon sequestration due to avoided forest conversion in the temperate and boreal regions in our measure of carbon sequestration due to changes in rotation length.

We then approximated the contribution of each individual activity to the total mitigation level. We acknowledge that important interactions between the activities are unaccounted through this approach. For instance, the area of afforestation is likely underestimated when considered alone because changing rotations and management both increase the productivity of forests, making forests more valuable, and incentivizing additional afforestation. However, as we used this approach to disaggregate our total sequestration estimate across activities, the error is likely to be small. This approach is a substantial improvement over studies which

estimate total mitigation based on the summation of individual mitigation activities without any consideration of their interactions.

**Country disaggregation of carbon sequestration data**. To disaggregate carbon to the national level, we started with the regional estimates of carbon fluxes by activity. For rotations, we disaggregated regional sequestration using observed industrial roundwood harvests. Changes in rotations can only occur on forests that are harvested for timber purposes, and if a country has a large share of total industrial wood harvest in a region, we assumed it is responsible for a proportional share of the change due to changes in rotations. Using data from FAOSTAT (2019), we determined each countries' proportional contribution to its regional production of industrial roundwood, and allocated the regional estimate using that proportion.

For management, we used the area of planted trees found in UN FAO (2015) to allocate the carbon gains from management. This approach makes sense given that management can only occur in places where trees are planted. We calculated the proportion of a region's total planted tree area in a given country, and use that proportion to allocate carbon gains. There is some potential that this underestimates carbon allocation due to management in some countries if there is large potential to shift existing forests that are not managed into replanted forests.

For land use change, comprising avoided forest conversion and afforestation/reforestation, we used data on forest change from 2005-2015 from UN FAO (2015). We calculated the absolute value of annual forest change over that time period for each country and for the region. We then allocate the region's carbon flux due to land use change to each country based on the proportion of the region's land use change observed in that country.

**GTM isolation of private land management costs**. In addition to total investment costs, we also calculated costs borne by land managers. Private investment costs represent changes in producer's surplus, a common economic welfare measure representing economic rents allocated to producers. In any given decadal simulation step, producer's surplus is measured as the difference between the total value of production and all costs. Specifically, we disaggregate GTM objective function components into the following cost and value elements:

(1) Total value—the product of endogenous prices and quantities for both sawtimber and pulpwood.
(2) Harvest costs—total costs of harvesting and distributing forest biomass to sawtimber and pulpwood markets.
(3) Management investment costs—costs accrued for increasing forest productivity through management, including replacement of naturally regenerated forests with plantation forests, thinning, fertilizers, and other silvicultural practices used to enhance productivity.
(4) Land rental costs—costs captured by rental functions that represent the opportunity costs of keeping land in forestry at the margin. Keeping land in forestry (e.g., avoided deforestation) or increasing forest area at the extensive margin will yield higher rental payments. These rents represent foregone economic opportunities if the land were used for alternative purposes such as agricultural production.

Supplementary Figure 2 provides a conceptual depiction of this approach. This figure distinguishes between costs borne by land managers and the total costs of mitigation, or the product of the exogenously defined mitigation price ($P^*$) and the endogenously determined mitigation quantity ($Q^*$). More information on the GTM objective function and cost structure can be found in refs. [19,20]. Isolating cost elements this way provides a unique perspective on potential investment costs needed to support different levels of mitigation. The producer cost changes between scenarios are a proxy for the minimum investment needed to support mitigation at a given quantity. This is compared to total mitigation costs, which also reflects consumer surplus losses and wealth transfers from entities supporting the mitigation action to land managers.

## Data availability
The data supporting the findings of this study are available from the Global Timber Model repository at https://u.osu.edu/forest/

## Code availability
The code supporting the findings of this study are available from the Global Timber Model repository at https://u.osu.edu/forest/

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

## Acknowledgements

The views expressed in this article are those of the authors and do not necessarily represent the views or policies of the US Environmental Protection Agency.

## Author contributions

K.G.A., J.S.B., B.L.S., and C.M.W. designed the study, with contributions from A.D., S.B.O., S.R., and A.B. B.L.S. ran the GTM model, and K.G.A., J.S.B., B.L.S., C.M.W., A.D., S.B.O., S.R., and A.B. analyzed and interpreted the data. K.G.A, J.S.B., B.L.S., C.M.W., A.D., and A.B wrote the manuscript.

## Competing interests

The authors declare no competing interests.
