## [Peer Review File · Nature Communications]

Reviewers' comments:

Reviewer #1 (Remarks to the Author):

In "How much will global forest carbon sequestration cost?" Baker and co-workers use GTM, a dynamic economic model to estimate the cost of C-mitigation in the forestry sector. Although this is a timely question and an evidence-based answer is much needed, this particular study and/or the manuscript presenting the study could not convince. Below I list issues/questions that, in my opinion, need to be addressed before such a study should be considered for publication in Nature Communications or another journal.

(1) How does this study go beyond state-of-the-art? I had to read the material to understand the key messages better. The main text should be rewritten such that the messages can be understood and appreciated before reading the more detailed method description. The manuscript could begin with a (short) description of the state-of-the-art and the main assumptions that are used in state-of-the-art cost estimates. Which of these assumptions are no longer acceptable and were replaced by more realistic assumptions in GTM? Going beyond state-of-the-art should be an essential criterion to be published in Nature Communications.

(2) Why should we trust the GTM simulations? The classic approach to demonstrate model skill is by confronting the simulations with observations. The simulations start in the year 2010 so it seems feasible to compare the simulations for the period 2010-2020 against data of the last decade. Does the model show the correct responses between 2010 and 2020? One of the novelties of GTM seems to be its ability to account for changes in forest management. It should thus be shown that the simulated response of forest management to changes in wood price are indeed what has been observed.

(3) I noted and appreciated the effort to compare GTM to previous model studies but that approach assumes that the previous estimates are trustworthy. If so, why redoing them? If they are not trustworthy because some of their basic assumptions are too crude, why would they be valid benchmarks? In short, comparing GTM to other models gives confidence that the model builds on existing knowledge, insights and achievements but fails to show that the model has indeed the skills that are required to lend credibility to this study. The manuscript and its supplementary material should provide all elements to convince its readers to trust the outcomes of GTM better than the outcomes of previous simulations with other models.

(4) At present, the central piece of the manuscript is a sensitive analysis with different carbon prices and growth rates. Sensitivity analyses are one of the many tests that are expected in good modelling studies but they tell us more about the model than the process we want to study. Sensitivity analysis are considered routine. A comparison of policy scenarios (each of which would have a similar parameter sensitivity because they all use GTM) seems more likely to result in a high-impact study than a model sensitivity analysis.

(5) The sensitivity tests are poorly justified. Which assumptions underlie the 1 and 3 % growth rates? A growth rate suggests that the carbon price will increase which implies that the demand for mitigation will keep increasing which will only be the case if carbonization fails which in turn implies that the technological developments that could solve the climate crisis did not materialize. What would be the effect on the forest sector if technology becomes very successful and efficient and thus the C-sequestration demand decreases? In other words what is the outcome for negative growth rates? The authors may have good reasons to find this a ridiculous proposition but they should then justify why the only reasonable range for growth rates lies between 1 and 3% and is always positive?

(6) The result that higher C prices lead to more mitigation is not really novel but the table and figures all stress this issue. The novelty of the model is hardly reflected in the presentation of the

results. GTM accounts for forest management so a map with which management is expected in which regions would demonstrate the new capabilities of this model. GTM can account for policy incentives, this capability is not at all reflected in the results. The discussion mentions interactions between regions but given this is a unique feature of GTM it should be stressed. Which regions will compensate for each other? Which regions will take over the production from other regions, etc ...?

(7) I would have liked to see some results about the land sector: wood volume, wood prices (timber and pulp), production volume per region, forest area per region, management per region, import/export per region, mitigation per region and carbon sequestered per region. The model accounts for afforestation and reforestation, how much agricultural lands will need to be converted per region to reach the different mitigation potentials? How does each of the five abatement activities contribute to the total mitigation? How much mitigation will come from forest management and how much from afforestation. I noted that some of the management activities are not at all common at present, i.e., fertilization, the authors should mention how much fertilizer will be needed to achieve this mitigation potential and of course in how much N₂O emissions this will result. I noted the use of the CO₂ equivalent unit. Which GHG other than CO₂ are currently accounted for and thus justify the use of that unit?

(8) The approach assumes that the land manager gets a price for a forest service, in this study carbon sequestration. This is only feasible if society agrees that there is a climate crisis and that they are willing to pay for it. Society is also facing other crises: a biodiversity crises and the clean-energy supply crises. If society is willing to pay to mitigate climate changes, chances are fair that the same society would be willing to pay its land manager for possibly conflicting services and products, in this case, biodiversity protection and the production of biomass for energy production. This kind of considerations (or caveats of the current analysis) is absent from the discussion.

(9) If the land manager is getting paid for the C-sequestration of the forest (it seems that this study does not account for the C stored in wood products), will the land-manager have to pay society when the forest stops providing this service? An asymmetric system would be prone to fraud but how would a symmetric system affect the decision of the owners? Owners would have to make decision that minimize natural disturbances because such disturbance could then result in not only a loss of income but also a cost to pay to society.

Reviewer #2 (Remarks to the Author):

This is a very interesting study. Authors have looked into economic cost of large-scale forest sequestration expansion. According to study the global forest sector could sequester up to 5.8 Gt CO₂ yr⁻¹ at a cost of \$343-\$432 billion yr⁻¹.

This is a very interesting study. Authors have looked into economic cost of large-scale forest sequestration expansion. According to study the global forest sector could sequester up to 5.8 Gt CO₂ yr⁻¹ at a cost of \$343-\$432 billion yr⁻¹.

However, biological system is very much non linear, so the cost of mitigation. Authors have used Global Timber Model, a dynamic economic model to estimate cost of carbon sequestration by forest.

Authors have considered a range of assumed ambition in land-based mitigation investments by including six potential carbon prices (\$5, \$20, \$35, \$50, \$75, and \$100/tCO₂e) and both low and high assumed carbon price growth rates (1% and 3% per year).

Five abatement activities were considered in GTM; 1) Avoided deforestation, 2) Reducing harvesting of old growth forests, 3) Forest management activities (including fertilizing, changing seedling density, competition management, and species selection), 4) Increasing harvest rotations, and 5) Afforestation and reforestation (including intensively managed timber plantation establishment).

Overall study is clear but still confusing in terms of what kind forest we are talking about in the study. Intact Terrestrial forest, replanted Terrestrial forest, afforested terrestrial forest, managed terrestrial forest (I feel only this kind of forest were used for this study). What about coastal forest? Because if we are talking about regionally, Southeast Asian coastal forest can help in climate change mitigation for example 50% mangrove area cover is in south East Asian countries. And mangrove forest can store 5 x 8 times more carbon than tropical forests. Considering all type of forest in this study would be great. That's why authors have mentioned that tropical forest have more mitigation potential than northern and southern limit like Russia because of boreal forest. Including all kind of forest and also types (such as deforested, intact, managed, reforested and afforested) in this study would.

Also, deforestation was avoided however, deforestation is still happening. It is difficult to avoid complete deforestation and therefore using different deforestation rates from each region can change or increase the cost of mitigation. Same goes for reforestation and afforestation rate also. Also better to separate afforestation and reforestation word because definitions are different. For afforestation, land managers need to acquire new land or land where forest was not available before. What about degradation? Considering rate of all these terms will give better estimates of mitigation cost.

Therefore, this study can be improved by including all kind of scenario to generate better mitigation cost. Considering types of forest for each region will greatly generate robust mitigation cost.

Reviewers' comments:

Reviewer #1 (Remarks to the Author):

In “How much will global forest carbon sequestration cost?” Baker and co-workers use GTM, a dynamic economic model to estimate the cost of C-mitigation in the forestry sector. Although this is a timely question and an evidence-based answer is much needed, this particular study and/or the manuscript presenting the study could not convince. Below I list issues/questions that, in my opinion, need to be addressed before such a study should be considered for publication in Nature Communications or another journal.

(1) How does this study goes beyond state-of-the-art? I had to read the material to understand the key messages better. The main text should be rewritten such that the messages can be understood and appreciated before reading the more detailed method description. The manuscript could begin with a (short) description of the state-of-the-art and the main assumptions that are used in state-of-the-art cost estimates. Which of these assumptions are no longer acceptable and were replaced by more realistic assumptions in GTM? Going beyond state-of-the-art should be an essential criterion to be published in Nature Communications.

We appreciate this comment and have tried to be more specific and convincing about what differentiates this study from previous studies and to highlight our important contributions to the literature. This analysis is a novel application of an existing tool that provides new and meaningful information on the economic costs of global forest carbon sequestration. It improves on previous analysis in the following ways:

- While other mitigation analyses referenced in our manuscript report economic mitigation outcomes at the margin (e.g., X GtCO₂e year⁻¹ below a specific price threshold), no study that we are aware of has provided total economic cost of global mitigation in the forest sector, using a modeling framework that captures market and trade dynamics, intertemporal trade-offs, interactions between abatement actions, and spatial allocation in harvest patterns and mitigation activities, all of which have a strong influence on costs. Because proposed carbon policies fall outside of historical norms and will have strong implications for markets, it is critical to capture endogenous feedback between markets and land use/management as suggested in [1]. Some previous US studies have reported domestic agriculture and forestry sector economic welfare changes under similar GHG mitigation policy measures, and several studies have projected the costs and benefits of comprehensive climate policy as a proportion of total GDP [2]. However, we are unable to identify a comparable study that has calculated the economic costs of global carbon sequestration in the forest sector using a consistent methodology that captures the influence of these important dynamics. In addition, our approach allows for explicit quantification of mitigation costs and abatement levels at certain price thresholds and incentive growth rates, which is more intuitive for policy dialog (e.g., abatement XX GtCO₂e by 2050 will cost \$YY per year).*
- Second, most recent studies of mitigation potential from the land use sectors rely on marginal abatement cost estimates that are compiled from a variety of modeling studies conducted at various points in time using different modeling frameworks and baseline assumptions and are, in some cases, 10-15 years outdated [3-5]. Market conditions and land use trends have changed substantially over the last two decades, shifting the direct and indirect (opportunity) costs of*

mitigation investment. Given the relative importance of land sector abatement in climate stabilization pathways presented in [6], it is imperative to update cost estimates using updated data, contemporary baseline settings, and capturing other recent advances within individual modeling frameworks (e.g., recent GTM developments discussed in [1, 7], all using a globally consistent framework.

- *Third, our mitigation potential and cost estimates are global, and cover a wider range of mitigation options than the estimates provided in [8], which only covers the tropics and only considers abatement through avoided deforestation and reforestation. [8] uses an empirical model which reflects only differences in potential net returns from alternative land uses. This approach does not account for dynamic considerations and changing marginal user costs under different policy incentive structures. Dynamic considerations are critical for measuring the path of future costs over any time period because timber stocks grow, are managed, are harvested, and are affected by disturbance over time. Marginal user costs, an important concept in dynamic economic modeling, drive spatial- and temporal changes to different policy stimuli in GTM, as forest management and harvest levels respond to changing market conditions over across regions and over time. We present the marginal costs of carbon sequestration due to land use change, forest management, and changes in rotation ages for individual countries. These methods have not been presented before, and represent an important step forward in carbon sequestration cost analysis.*

Thus, this analysis presents a novel assessment of the economic costs of forest carbon sequestration by action and by country. We have added additional discussion in the Introduction highlighting the novelty of this application (Line 43-52, 106 – 117, 240 – 245, and 256 - 269).

(2) Why should we trust the GTM simulations? The classic approach to demonstrate model skill is by confronting the simulations with observations. The simulations start in the year 2010 so it seems feasible to compare the simulations for the period 2010-2020 against data of the last decade. Does the model show the correct responses between 2010 and 2020? One of the novelties of GTM seems to be its ability to account for changes in forest management. It should thus be shown that the simulated response of forest management to changes in wood price are indeed what has been observed.

To address the concern regarding model skill we compared GTM projections of regional harvest levels for industrial roundwood (total less fuelwood), pulpwood, and sawtimber over 2015 - 2020, to reported FAO harvest statistics over the same region (where FAO country level statistics are aggregated to GTM regions to facilitate comparison). We show that total roundwood harvests are within 12% of reported harvest statistics at the global scale, and that differences are relatively small across most productive timber regions (Table S1). To do this, we recalibrated initial conditions and updated the starting period in the simulations to reflect 2015 (or the decade between 2015 and 2024), rather than 2010 in the previous version. This starting point provides a more contemporary perspective on market conditions and is more in line with the starting point for the carbon pricing policies (which are initiated in 2020 in this analysis).

In addition, we have incorporated additional citations and discussion that address the concern regarding model performance, including [9] and [10]. [9] uses GTM to represent historical forest management and land use starting from 1900, and provide an approximation of observed forest land use and harvest levels using the same modeling framework and input assumptions in the modeling framework used for this analysis. This study additionally highlights the importance of management's contributions to the

historic evolution of forest carbon stocks. The latter citation addresses uncertainty in future model projections, specifically related to parametric uncertainty. That paper demonstrates the relative stability in GTM projections under uncertainty in future forest yield growth functions and economic parameters governing the opportunity costs of forestland. These projections, and the relatively small uncertainty bounds associated with forest land use and carbon stocks, increase confidence that GTM projections are stable relative to observed forest land use, harvests, and market conditions, and thus provide a forward looking validation of the model's ability to reflect the global forest sector.

This discussion is presented in a new Methods section 'GTM Validation' L 376 - 395.

(3) I noted and appreciated the effort to compare GTM to previous model studies but that approach assumes that the previous estimates are trustworthy. If so, why redoing them? If they are not trustworthy because some of their basic assumptions are too crude, why would they be valid benchmarks? In short, comparing GTM to other models gives confidence that the model builds on existing knowledge, insights and achievements but fails to show that the model has indeed the skills that are required to lend credibility to this study. The manuscript and its supplementary material should provide all elements to convince its readers to trust the outcomes of GTM better than the outcomes of previous simulations with other models.

Per our responses to comment #1, we have added additional discussion in the main text emphasizing the primary contributions of our manuscript. We clarify the importance of (1) structural dynamic considerations, (2) up to date and consistent cost estimates, (3) a global-scale perspective, (4) the ability to represent forest management and harvest, when modeling mitigation potential from the forestry system, (5) and the ability to calculate costs by activity globally and at the national level. This is further emphasized Line 43-52, 106 – 117, and 240 – 245.

(4) At present, the central piece of the manuscript is a sensitive analysis with different carbon prices and growth rates. Sensitivity analyses are one of the many test that are expected in good modelling studies but they tell us more about the model than the process we want to study. Sensitivity analysis are considered routine. A comparison of policy scenario's (each of which would have a similar parameter sensitivity because the all use GTM) seems more likely to result in a high-impact study then a model sensitivity analysis.

We appreciate this comment but believe our approach for conducting sensitivity analysis around the carbon price starting point and growth rate is appropriate for generate marginal and total cost estimates of carbon sequestration policies of various scale. Such marginal cost estimates have long been useful and have provided critical information to guide policy dialog and mitigation investments (e.g., [8, 11-13]. Of course, developing policy scenarios, or socioeconomic narratives such as the shared socioeconomic pathways (SSPs), and measuring costs or sequestration potential along those is also useful, but this is a different exercise and provides different information. Our current approach allows for direct quantification of costs and we evaluate a broad range of price scenarios and price paths representing different levels of policy ambition over time, offering information on both the costs of achieving near-term commitments and scenarios that are more representative of long-term climate stabilization pathways.

(5) The sensitivity test are poorly justified. Which assumptions underlie the 1 and 3 % growth rates? A growth rate suggests that the carbon price will increase which implies that the demand for mitigation will keep increasing which will only be the case if carbonization fails which in turn implies that the technological developments that could solve the climate crisis did not materialize. What would be the effect on the forest sector if technology becomes very successful and efficient and thus the C-sequestration demand decreases? In other words what is the outcome for negative growth rates? The authors may have good reasons to find this a ridiculous proposition but they should then justify why the only reasonable range for growth rates lies between 1 and 3% and is always positive?

We agree that there is a plausible scenario in which carbon price rise and then fall. Indeed, the European ETS and the US SO₂ markets are excellent examples of just that phenomenon, but both of those are markets for flow pollutants (the ETS had a focus on a single commitment period, so CO₂ targets were a flow target). Theoretically, carbon is a stock pollutant, and if one believes that the damages from carbon stocks in the atmosphere are at least as great as the social cost of carbon estimated by [14], then the price of carbon will not go down until the stock of carbon goes down. The IPCC Special Report on 1.5C Warming showed stocks falling in some scenarios to achieve a maximum temperature increase of 1.5C. But these scenarios also implied extremely high carbon prices were required to achieve such a goal. Given that our model is a dynamic model that treats intertemporal considerations the same as the DICE model, we have chosen the prices and growth rates to lie within the range of prices and growth rates observed in various scenarios from the DICE model. Creating a separate scenario that assumes carbon prices and carbon stocks decline is interesting but is beyond the scope of this paper.

(6) The result that higher C prices lead to more mitigation is not really novel but the table and figures all stress this issue. The novelty of the model is hardly reflected in the presentation of the results. GTM accounts for forest management so a map with which management is expected in which regions would demonstrate the new capabilities of this model. GTM can account for policy incentives, this capability is not at all reflected in the results. The discussion mentions interactions between regions but given this is a unique feature of GTM it should be stressed. Which regions will compensate for each other? Which regions will take over the production from other regions, etc ...?

In response to this comment we have refined our analysis to disaggregate the mitigation results by activity, including avoided deforestation and afforestation, rotation extension, and forest management; and to disaggregate the results by country. This allows us to report region-level mitigation projections from the range of activities represented in GTM, at the country-level. These results are novel in that they show long-term mitigation potential for more countries than previous studies focused on the tropics (e.g., [8]) and offer insight on management's potential role in boosting carbon stocks in the context of a market system, which is missing from the natural climate solutions literature (e.g., [3]). We documented this approach in the methods section L428 - 455, and present results in Figure 2 and L144 – 160 and L246-255.

(7) I would have liked to see some results about the land sector: wood volume, wood prices (timber and pulp), production volume per region, forest area per region, management per region, import/export per region, mitigation per region and carbon sequestered per region. The model accounts for afforestation and reforestation, how much agricultural lands will need to be converted per region to reach the different mitigation potentials? How does each of the five abatement activities contribute to the total mitigation? How much mitigation will come from forest management and how much from afforestation.

Per the response to comment # 6, we disaggregated our previous findings by mitigation activity and country, and now present these results in the main text (Figure 2, Figure 3). We have also added a supplemental table showing regional reallocation in harvest patterns in response to the carbon policies (Table S2).

We have added a table to the supplement that shows net changes in land use (avoided deforestation plus afforestation) by region and over time (Table S3). Afforestation is assumed to come primarily from agricultural lands, as land rental functions reflect the opportunity costs of moving land into forestry from other uses where the majority of historic land use exchanges have occurred (namely pasture and cropland). Afforestation increases with higher rental payment, indicating a higher proportion of productive agricultural lands will be displaced at higher mitigation prices. More discussion on land rental functions is provided in [10].

I noted that some of the management activities are not at all common at present, i.e., fertilization, the authors should mention how much fertilizer will be needed to achieve this mitigation potential and of course in how much N₂O emissions this will result. I noted the use of the CO₂ equivalent unit. Which GHG other than CO₂ are currently accounted for and thus justify the use of that unit?

GHG accounting in GTM captures carbon flux changes in aboveground forest carbon, wood products, and soils. We are not capturing non-CO₂ greenhouse gases from forest management, fertilization, or product processing, and have therefore corrected our units to reflect that we are only accounting for CO₂. Fertilizers are used only on the most intensively managed forests in the model, which comprises 96 million ha of intensively managed plantations globally initially. In the baseline the area grows to 140 million ha by 2120. These plantations are assumed to receive 120 kg/ha N and 35 kg/ha P one time during their rotation (10-30 years depending on location and species). We do not track the N emission implications of this management activity, although we recognize its importance.

(8) The approach assumes that the land manager gets a price for a forest service, in this study carbon sequestration. This is only feasible if society agrees that there is a climate crisis and that they are willing to pay for it. Society is also facing other crises: a biodiversity crises and the clean-energy supply crises. If society is willing to pay to mitigate climate changes, chances are fair that the same society would be willing to pay its land manager for possibly conflicting services and products, in this case, biodiversity protection and the production of biomass for energy production. This kind of considerations (or caveats of the current analysis) is absent from the discussion.

We agree that society would need to demonstrate a greater willingness to commit to decarbonization policies for payment programs such as those applied in this manuscript could become a reality. We also agree that biodiversity is a concern, and that forest loss and even forest management can cause biodiversity loss. There may be tradeoffs associated between carbon sequestration and biodiversity objectives. In addition, there may be complementarities between carbon sequestration and biomass energy policies (as noted in for example in [15, 16]). We have not tried to quantify those tradeoffs and complementarities, which are potentially important but are outside the scope of this paper. We do discuss the potential implications of the role of afforestation/reforestation and intensively managed plantations in mitigation programs, with respect to biodiversity and ecosystem regulating services (L202 – 206).

(9) If the land manager is getting paid for the C-sequestration of the forest (it seems that this study does not account for the C stored in wood products), will the land-manager have to pay society when the forest stops providing this service? An asymmetric system would be prone to fraud but how would a symmetric system affect the decision of the owners? Owners would have to make decision that minimize natural disturbances because such disturbance could then result in not only a loss of income but also a cost to pay to society.

We apologize for the confusion about how carbon sequestration is incentivized in this paper. We use the carbon rent approach described in [17] and [16]. The carbon rental approach rents carbon in forest stocks at a rental "price" consistent in net present value terms with the path of current and future carbon prices. At harvest time, carbon permanently stored in wood products is paid the carbon price at that time. Carbon released into the atmosphere is released without payment or tax. This approach fully accounts for the value of the carbon interaction between forests and the atmosphere, as discussed in the two citations above. The rental approach is efficiently equivalent to the tax and subsidy approach [18]. By efficiently equivalent we mean that both approaches correctly value the exchange in carbon between the atmosphere and forested ecosystems, including harvested wood products, and with the same set of prices, they both lead to the exact same result in our model. The rental approach is easier to implement in the model, however, and is thus what we use.

It is interesting to note that the rental approach does not penalize people or countries for their past carbon storage. So for instance under a tax and subsidy scheme, an HFLD (high forest low deforestation country) would receive no payment (because stocks are stable) and would be penalized if deforestation occurs. Under the rental approach, they are not penalized in any case, and they receive a payment for any deforestation they avoid. This approach is more consistent with current policy. We have improved explanation in the text L 341 – 374.

Reviewer #2 (Remarks to the Author):

This is a very interesting study. Authors have looked into economic cost of large-scale forest sequestration expansion. According to study the global forest sector could sequester up to 5.8 Gt CO₂ yr⁻¹ at a cost of \$343-\$432 billion yr⁻¹. However, biological system is very much non-linear, so the cost of mitigation.

We agree the biological system is nonlinear, and note that this is accounted for in our model. For instance, we treat forest growth dynamics with a logistical growth function. Furthermore, the decision to replant or manage forests maximizes the net present value of a stand, and thus is an inherently non-linear decision. As a result of the non-linearity we have modeled, our resulting marginal cost functions are non-linear. One non-linearity that is not addressed here is the role of climate change, which could have widespread consequences for forest growth. We have acknowledged that issue in the paper L 89 – 91, but addressing it is beyond the scope of this paper.

Authors have used Global Timber Model, a dynamic economic model to estimate cost of carbon sequestration by forest. Authors have considered a range of assumed ambition in land-based mitigation investments by including six potential carbon prices (\$5, \$20, \$35, \$50, \$75, and \$100/tCO₂e) and both low and high assumed carbon price growth rates (1% and 3% per year). Five abatement activities were considered in GTM; 1) Avoided deforestation, 2) Reducing harvesting of old growth forests, 3) Forest management activities (including fertilizing, changing seedling density, competition management, and

species selection), 4) Increasing harvest rotations, and 5) Afforestation and reforestation (including intensively managed timber plantation establishment).

Overall study is clear but still confusing in terms of what kind forest we are talking about in the study. Intact Terrestrial forest, replanted Terrestrial forest, afforested terrestrial forest, managed terrestrial forest (I feel only this kind of forest were used for this study). What about coastal forest? Because if we are talking about regionally, Southeast Asian coastal forest can help in climate change mitigation for example 50% mangrove area cover is in south East Asian countries. And mangrove forest can store 5 x 8 times more carbon than tropical forests. Considering all type of forest in this study would be great. That's why authors have mentioned that tropical forest have more mitigation potential than northern and southern limit like Russia because of boreal forest. Including all kind of forest and also types (such as deforested, intact, managed, reforested and afforested) in this study would.

Thanks for this point. To address this comment we provide more information L 300 – 320 regarding the forest types included in our model. We have included 275 total forest types from around the world. These forest types all have different growth functions and different levels of opportunity cost as well as other costs. However, we do not differentiate mangrove forests, although these are a notably valuable forest type in many regions of the world.

Also, deforestation was avoided however, deforestation is still happening. It is difficult to avoid complete deforestation and therefore using different deforestation rates from each region can change or increase the cost of mitigation. Same goes for reforestation and afforestation rate also. Also better to separate afforestation and reforestation word because definitions are different. For afforestation, land managers need to acquire new land or land where forest was not available before. What about degradation? Considering rate of all these terms will give better estimates of mitigation cost.

We have separated avoided deforestation and afforestation as well as forest management in the analysis and explicitly included marginal cost functions for each. We agree that it is difficult to design policies to address avoiding deforestation, and we have focused solely on the economic question about what the costs would be if actors (i.e. those deforesting) always responded perfectly to economic instruments. Of course, in reality, not all (or even most) people will respond this way.

Therefore, this study can be improved by including all kind of scenario to generate better mitigation cost. Considering types of forest for each region will greatly generate robust mitigation cost.

Thank you for this comment, we hope that the improvements made to differentiate mitigation quantities and costs by abatement activity and region will provide a more useful basis for policy evaluation and decision support.

References

1. Tian, X., et al., *Will U.S. Forests Continue to Be a Carbon Sink?* Land economics, 2018. **94**: p. 97-113.

2. van Vuuren, D.P., et al., *The costs of achieving climate targets and the sources of uncertainty*. Nature Climate Change, 2020. **10**(4): p. 329-334.
3. Griscom, B.W., et al., *Natural climate solutions*. Proceedings of the National Academy of Sciences, 2017. **114**(44): p. 11645.
4. Fargione, J.E., et al., *Natural climate solutions for the United States*. Science Advances, 2018. **4**(11): p. eaat1869.
5. Roe, S., et al., *Contribution of the land sector to a 1.5 °C world*. Nature Climate Change, 2019. **9**(11): p. 817-828.
6. IPCC, *Global Warming of 1.5 C An IPCC Special Report on the Impacts of Global Warming of 1.5 C Above Pre-Industrial Levels and Related Global Greenhouse Gas Emission Pathways, in the Context of Strengthening the Global Response to the Threat of Climate Change Sustainable Development, and Efforts to Eradicate Poverty*. 2019.
7. Kim, J.B., et al., *Assessing climate change impacts, benefits of mitigation, and uncertainties on major global forest regions under multiple socioeconomic and emissions scenarios*. Environmental Research Letters, 2017. **12**(4): p. 045001.
8. Busch, J., et al., *Potential for low-cost carbon dioxide removal through tropical reforestation*. Nature Climate Change, 2019. **9**(6): p. 463-466.
9. Mendelsohn, R. and B. Sohngen, *The Net Carbon Emissions from Historic Land Use and Land Use Change*. Journal of Forest Economics, 2019. **34**(3-4): p. 263-283.
10. Sohngen, B., et al., *The Influence of Parametric Uncertainty on Projections of Forest Land Use, Carbon, and Markets*. Journal of Forest Economics, 2019. **34**(1-2): p. 129-158.
11. IPCC, *Summary for Policymakers, in Climate Change 2007: The Physical Science Basis. Contribution of Working Group I to the Fourth Assessment Report of the Intergovernmental Panel on Climate Change*, S. Solomon, et al., Editors. 2007: New York, Ny, USA.
12. IPCC, *Synthesis Report. Contribution of Working Groups I, II and III to the Fifth Assessment Report of the Intergovernmental Panel on Climate Change*. 2014, R.K. Pachauri and L.A. Meyer (eds.): Geneva, Switzerland.
13. Kindermann, G., et al., *Global cost estimates of reducing carbon emissions through avoided deforestation*. Proceedings of the National Academy of Sciences, 2008. **105**(30): p. 10302-10307.
14. Nordhaus, W.D., *Revisiting the social cost of carbon*. Proceedings of the National Academy of Sciences, 2017: p. 201609244.
15. Favero, A., R. Mendelsohn, and B. Sohngen, *Using forests for climate mitigation: sequester carbon or produce woody biomass?* Climatic Change, 2017. **144**(2): p. 195-206.
16. Favero, A., A. Daigneault, and B. Sohngen, *Forests: Carbon sequestration, biomass energy, or both?* Science Advances, 2020. **6**(13): p. eaay6792.
17. Sohngen, B. and R. Mendelsohn, *An Optimal Control Model of Forest Carbon Sequestration*. American Journal of Agricultural Economics, 2003. **85**(2): p. 448-457.
18. van Kooten, G.C., C.S. Binkley, and G. Delcourt, *Effect of Carbon Taxes and Subsidies on Optimal Forest Rotation Age and Supply of Carbon Services*. American Journal of Agricultural Economics, 1995. **77**(2): p. 365-374.

REVIEWERS' COMMENTS:

Reviewer #2 (Remarks to the Author):

Authors have addressed review comments appropriately.

Market conditions and land use trends have changed substantially over the last two decades. Authors need to add few sentences how and what kind changes occurred? And also do authors think in future these changes will be keep happening if yes how it will change the scenario.

It would be better to add few sentences about what kind of aspects needed to included in the future study. Such as including CH₄ and N₂O or halocarbons emissions from the forest. There may be tradeoffs associated between carbon sequestration and biodiversity objectives. In addition, there may be complementarities between carbon sequestration and biomass energy policies. This could be also included in the future studies.

Response to Review on “The economic costs of planting, preserving, and managing the world's forests to mitigate climate change”.

Thank you for the constructive feedback. Please see our responses below:

Reviewer #2 (Remarks to the Author):

Authors have addressed review comments appropriately.

Market conditions and land use trends have changed substantially over the last two decades. Authors need to add few sentences how and what kind changes occurred? And also do authors think in future these changes will be keep happening if yes how it will change the scenario.

We acknowledge that modelling during a dynamic period of change is challenging. To address this, we added a section in the manuscript L #-# : “Forest loss in temperate and boreal regions has slowed, and even reversed, over the past several decades, due to demographic transitions, changes in consumption patterns, and greater investments in forest management. Tropical forest loss is likely to decline in the future due to slowing population growth, shifts in consumer preferences, and stricter land use policy (Daigneault et al., 2019). These land use patterns are captured by our reference scenario projections, which is calibrated to FAO forest land use for different regions in the model and through land rental functions that are adjusted over time to reflect higher opportunity costs of both agricultural and forest land.”

Daigneault, A., Johnston, C., Kurosuo, A., Baker, J. S., Forsell, N., Prestemon, J., and Abt., R. (2019). Developing Detailed Shared Socioeconomic Pathway (SSP) Narratives for the Global Forest Sector. *Journal of Forest Economics*. 34(1-2), 7-45.

It would be better to add few sentences about what kind of aspects needed to included in the future study. Such as including CH₄ and N₂O or halocarbons emissions from the forest.

There may be tradeoffs associated between carbon sequestration and biodiversity objectives. In addition, there may be complementarities between carbon sequestration and biomass energy policies. This could be also included in the future studies.

To acknowledge these future areas of research we added a comment to the first paragraph of the Discussion (L248 – 251): “Future research that improves and expands on this assessment by considering, for example, non GHG aspects of climate forcing, non-CO₂ fluxes, the potential impacts of climate change on forest growth and productivity, and the influence of bioenergy policies, will further refine our understanding of the role of forests in global climate stabilization.” We also acknowledge the need for further investigation of the potential impacts on biodiversity L217 – 218.